

# Climate Change impacts the *Water Highway* project in Morocco

Nabil El Moçayd[1], Suchul Kang[2], and Elfatih A. B. Eltahir[2]

[1]International Water Research Institute, University Mohammed 6 Polytechnic, Lot 660 Hay Moulay Rachid Benguerir 43150, Morocco
[2]Ralph M. Parsons Laboratory, Department of Civil and Environmental Engineering, Massachusetts Institute of Technology, Cambridge, MA 02139, USA

**Correspondence:** Nabil El Noçayd (nabil.elmocayd@um6p.ma)

**Abstract.** The hydrology of Morocco is characterized by a significant spatial variability. Precipitation follows a sharp gradient decreasing from the North to the South. In order to redistribute water, a project is proposed to transfer $860$ million $m^3$ per year from the wet north to the arid southern regions, *{Water Highway}*. The present study aims to address the viability of the project including the effects of climate change in the watersheds located in the North. We perform Regional Climate Model

5   (RCMs) simulations over the study region using boundary conditions from five different global circulation models (GCMs) and following two emissions scenarios RCP4.5 (with mitigation) and RCP8.5 (business as usual). The impact on precipitation is assessed and the decrease of available water quantity is estimated. Under RCP 8.5 the project is likely unfeasible. However under the RCP 4.5 a rescaled version of this project may be feasible depending on how much water is allocated to satisfy the local water demand.

## 1   Introduction

In many regions in the world, water scarcity is a critical issue that should be seriously addressed by stakeholders. Greve et al. (2018) define water scarcity as the ratio between the natural supply and the local demand in water. As a result, water scarcity depends mainly on two factors: hydro-climatology and socio-economics (Wada et al. (2014)). While a study shows that Mo-

15   rocco is not much threatened by the increase of population (Boke-Olén et al. (2017)). Several studies suggest that arid and semi arid regions like Morocco should develop new policy in order to adapt to climate change (Kahil et al. (2015); Batisani and Yarnal (2010); Lioubimtseva and Henebry (2009)). As reported by Greve et al. (2018), different policies could be adopted, including investments in structures for transferring water from a basin to another.

Hydrology in Morocco is characterized by strong spatial and temporal variability (Driouech et al. (2010); Tramblay et al.

20   (2013)). This uneven distribution of water has led the Moroccan government to develop hydraulic infrastructure to improve the local management of water resources in the Kingdom. In 1967, the King Hassan II launched a project to irrigate 1 million hectares by the year 2000. More precisely, the plan was to build a dam every year. The outcomes of this strategy helped to





achieve food security, increase cash crops production for export and improve the social and economic level of local farmers. More implicitly the construction of these infrastructures has helped communities to adapt to climate variability. Previous studies (Doukkali (2005)) showed that although these efforts resulted in a significant optimization of water resources management in Morocco, they remain insufficient to insure water and food security. A new policy is needed to complement all these efforts. By

2020 only the watersheds in the north of Morocco will have excess of water compared to the demand (Minoia and Brusarosco (2006)). The proposed North-South Water Transfer project (*Water Highway*) aims to supply water in the arid southern regions from the watersheds in the north (see Figure 3). The projected quantity of water to be transfered by this project is 860 million $m^3$ per year (Water-Office (2017)).

Other countries adopted similar policy in order to alleviate water scarcity problems. In China (Gu et al. (2012)), water is trans-

ferred from the tropical southern region to the very dense, industrialized and arid north, and the project is named South-North Water Transfer Project (SNWTP). The goal of this project is to transfer $44.5\ km^3$ each year from the Yangtze River basin. Although the economic benefit of the project is appreciated in the region (Feng et al. (2007)), yet it has raised some controversies (Berkoff (2003)). First transferring water in open channels increases risk of pollution (Tang et al. (2014)). Traffic accidents specially involving trucks represent a threat to water quality. Moreover, the risk of salinization of both basins (due to the trans-

fer for the receiving one and due to the sea water intrusion in the other basin) is highlighted by (Zhang (2009)). Therefore, significant efforts are to be carried in order to improve watersheds management (Zhang et al. (2009)). *Water Highway* could also lead to a dramatic increase of water cost. These concerns have been raised in the case of SNWTP by two studies (Wei et al. (2010); Yang and Zehnder (2005)). In fact, the project relies significantly on water availability in the south of China. And as it has been suggested by (Liu and Zheng (2002); Li et al. (2015)), water quantity will decrease in the donor watersheds due

to global climate change. This decrease will not affect the viability of the project in the Chinese case nonetheless. In contrast it is not clear for the Moroccan case, if the change in precipitation would not change the viability of the project. Therefore the goal of the present study is to assess the viability of the *Water Highway* including the climate change impact.

Global change is impacting the African climate. Indeed, many studies showed that regional climate of Africa is likely to change during the upcoming decades (Sowers et al. (2011)). This change is likely to have considerable socio-economical and environ-

mental impacts. A recent study that used the IPCC AR4 ($4^{th}$ Assessment Report of the Intergovernmental Panel on Climate Change) emissions scenarios to predict climate change in Africa, reported that the northern part is more likely to suffer from an increase in temperature and a decrease of precipitation in amount and frequency (Vizy and Cook (2012)). As a result, North Africa will be vulnerable to climate change (Döll (2009)).

In last decades, Morocco has built one of the strongest economies in Africa (101 billion US Dollar in GDP as reported by

the World Bank "https://data.worldbank.org/country/morocco"). This economy, however efficient compared to other developing countries, depends significantly on water availability (Agriculture represents 15% of the GDP). Water imbalance between demand and supply is growing (Hellegers et al. (2013)). In Morocco, water demand is affecting water scarcity more intensely than climate change (Vörösmarty et al. (2000)). On one hand, the urban population is expected to be increasing by about 270 thousand inhabitant per year due to urbanization, as reported by the Moroccan High Commission for Planning Website

(http://www.hcp.ma/). On the other hand, the improved irrigation schemes has certainly intensified cash crops production in



Morocco, but has also increased water demand. In Ward and Pulido-Velazquez (2008), for instance authors show that the increase of efficiency increases water use. Furthermore, climate change projections predict that Morocco will experience a dry future (Giorgi and Lionello (2008); Kang et al. (2019)) and a decrease in precipitation ( Patricola and Cook (2010)). For all these reasons among others, the country is increasingly vulnerable to global change.

The purpose of the present work is to study the viability of the North-South Water Transfer project, considering the impact of climate change. More precisely, the study aims to assess the effect of climate change on water availability in the northern basins. In order to evaluate climate change impact on this project, multi-model climate simulations were performed. Forcing boundary conditions from five GCMs (MPI, GFDL, IPSL, CSM, ACCESS ) were considered in order to quantify the uncertainty in future climate projections (Tebaldi and Knutti (2007)). The simulations performed using MRCM (MIT Regional

Climate Model) simulates accurately the global change impact on precipitation and temperature (Eltahir et al. (2013)). The impact on runoff and therefore on water availability is computed using runoff elasticity approach (Sankarasubramanian and Vogel (2003); Tang and Lettenmaier (2012)) based on available data. Such methodology can be used to evaluate the impact of climate change on water availability in regions where hydrological modelling may not be feasible.

The article is presented as follows: The second section describes the North-South Water transfer project in Morocco, the northern region that ought to supply water to the arid regions in the South, and the available datasets needed to perform the analysis. The third section is dedicated to present methodologies used in order to assess the expected loss of water in the northern watersheds. Finally the fourth section presents the results. Conclusion and perspectives are described in section 5

## 2   North-South Water transfer project

### 20   2.1   Water distribution in Morocco

Morocco is located in the North-West of Africa. Given this geographical position, the climate is characterized by a strong spatial variability. Indeed the southern part of the country is affected by the extreme dry climate of the Sahara desert, and the north-western part benefits from the moderate climate of Mediterranean sea and Atlantic ocean. In addition, the topography of Morocco varies also in space. As the coastal region is flat while, in the north and the east, the country's topography is

shaped by the Rif mountains and Atlas mountains respectively. Climate therefore can vary from sub-humid to arid over the country (Born et al. (2008)). As a result, the precipitation over Morocco varies strongly in space. Figure 1 displays the annual mean precipitation over Morocco using TRMM (Tropical Rainfall Measuring Mission) data (Huffman and Bolvin (2013)). The annual precipitation reaches up to $800 \ mm/yr$ in the north, while the south barely receives $100 \ mm/yr$. Moreover, the annual mean precipitation in Morocco is $240 \ mm/yr$ which lead to a strong hydrological stress (see Figure 2). As a result, only the

extreme north-west part of the country enjoys excess in water supply.





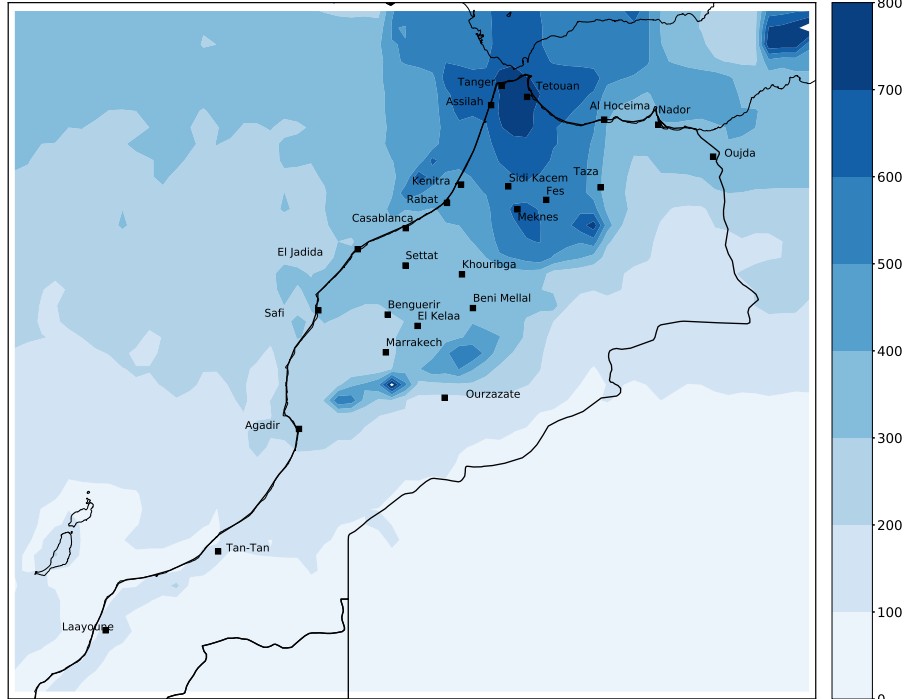

**Figure 1.** Annual precipitation (mm) in Morocco using TRMM data

## 2.2 The general features of the project

As reported by the Moroccan department in charge of water, plans are ready to launch the project of water transfer from north to south Water-Office (2017). The water balance analysis prepared in the studies carried out by the Moroccan Department of Water shows that southern basins notably Tensift and Oum Er Rbia are in deficit. On the other hand the northern basins discharge

5   excess water to the sea. A transfer from the northern basins, which are much better equipped with hydraulic structures, would relieve these deficits thus consolidating national integration in water management. The preliminary pre-feasibility studies conducted by the Department of Water has shown that roughly 860 million $m^3$ could be transferred on average from basins of Oued Laou, Loukkos and Mjara (See Figure 4) to the basins of Bouregreg, Oum Er Rbia and Tensift (See Figure 3). The project implementation schedule as well as the institutional and financial setup will be defined by the studies in progress by

10  both concerned ministerial departments namely water and agriculture.

The project, following three phases as described in Table 1, will involve the construction of 9 pumping stations, 2 new dams and 500 $km$ of channels and pipes to transfer water from Oued Laou, Loukkos and Sebou to Al Massira Dam.



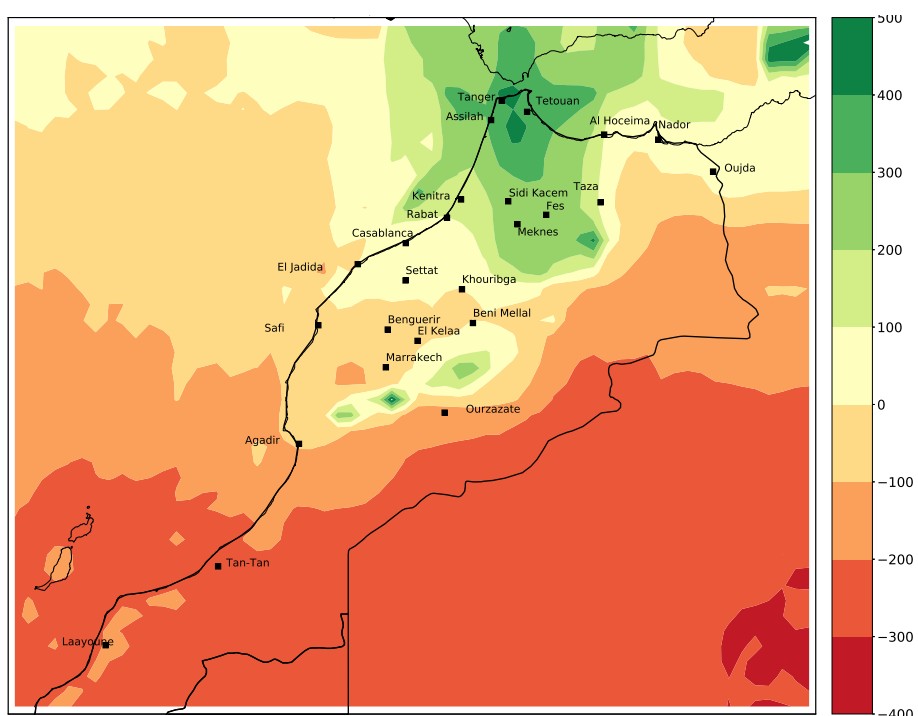

**Figure 2.** Precipitation departure from the spatial annual average in (mm)

| Phase | Start Point | End point | Water Amount (million $m^3$/yr) | Cost (billion MAD) | Cost (billion US $) |
|---|---|---|---|---|---|
| Phase 1 | Kodiat al Borna (near Al Wahda Dam) | Al Massira Dam | 392 | 16.3 | 1.63 |
| Phase 2 | Beni Mansour (Oued Laou) | Al Makhazine Dam | 726=334 + 392 | 13 | 1.3 |
| Phase 3 | Oued Sebou | Phase 1 pipeline route | 860= 726 + 134 | 1.7 | 0.17 |
| Project | Oued Sebou, Loukkos, Oued Laou | Al Massira Dam | 860 | 40 | 4 |

**Table 1.** The 3 phases for the project





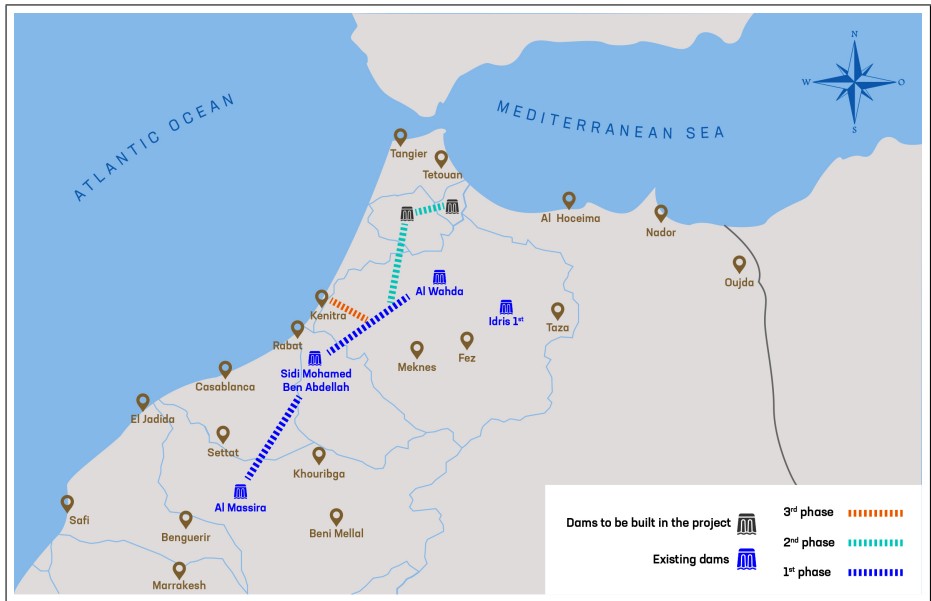

**Figure 3.** The planned Water Transfer Project

## 2.3 Description of watersheds of interest

The purpose of this study is to analyse the effect of climate change on the project. In order to address the viability of the project, global climate change impact on the hydrology of the donor watersheds should be assessed. Figure 4 presents the three watersheds that will supply the project, namely Loukkos, Oued Laou and Mjara (presented in a red contour). The latter

5 is a sub-watershed of Sebou watershed, supplying most of the Sebou water. The three water basins are located in the northern region of Morocco, where the precipitation levels are relatively high. There is a spatial gradient of elevation from the north-east to the south-west due to the presence of Rif Mountains (see Figure 5).

## 3 Datasets and methodologies

### 3.1 Datasets

In order to perform this study, we accessed several datasets. Table 2 summarizes the available data. Each data is available at a monthly time step.

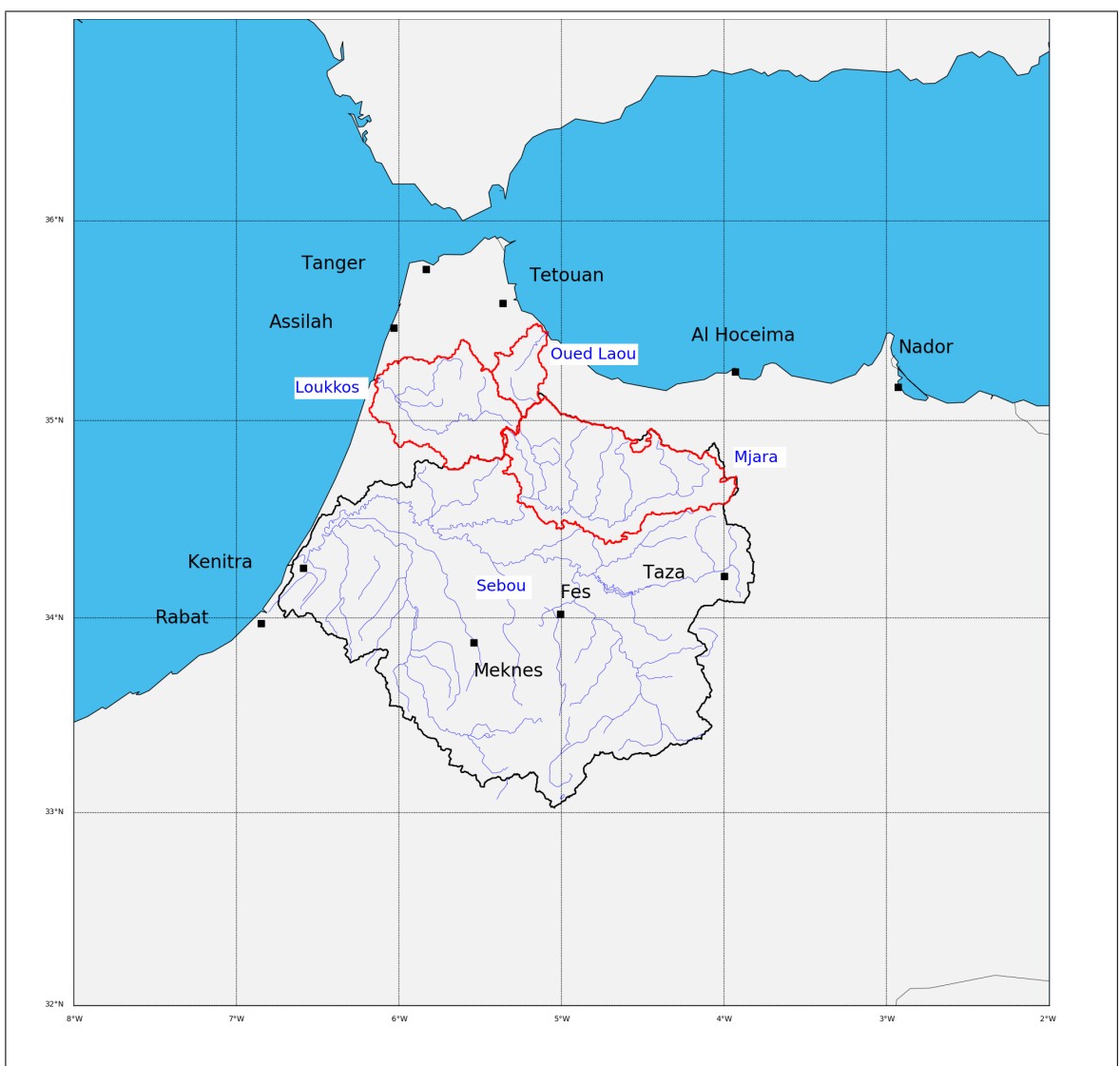

**Figure 4.** Watersheds of interest

### 3.1.1 Spatial Data

Spatial data helps to understand the temporal as well as the spatial variability of precipitation. For the present case we use the data from the space mission TRMM (Tropical rainfall measurement mission) (Huffman and Bolvin (2013)). We downloaded remote sensing data only for the region of Morocco and extracted it for the three watersheds considered in this analysis.





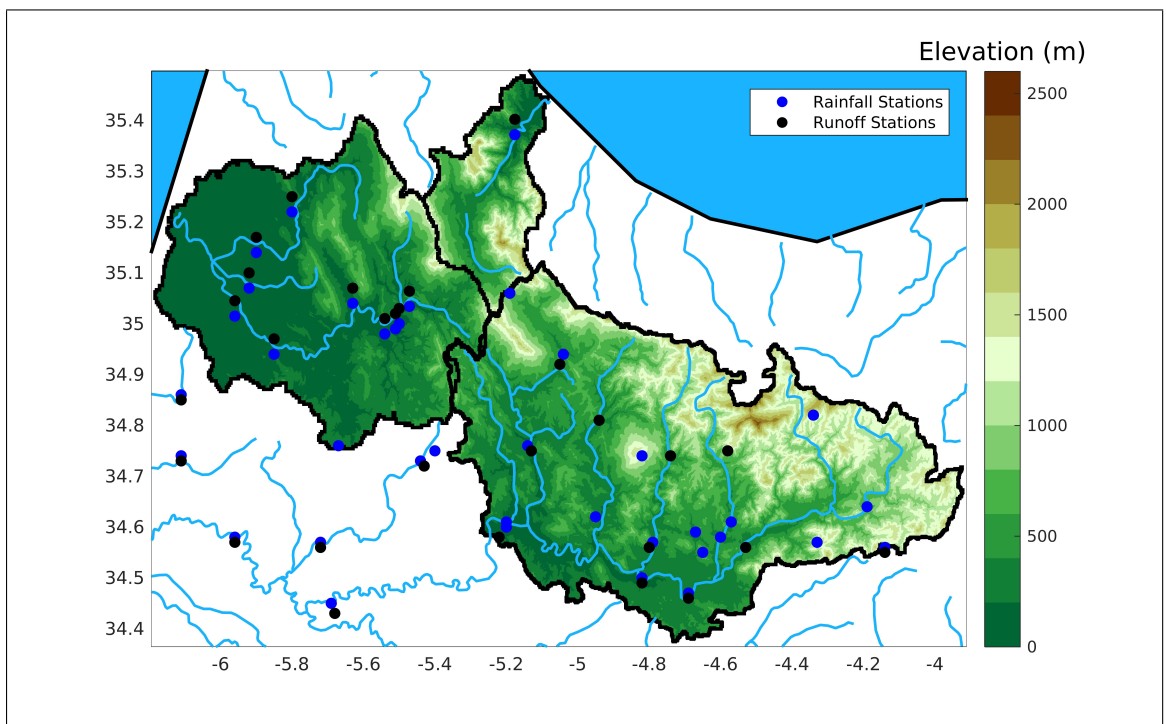

**Figure 5.** Mapping of the elevation and the position of gauge stations.

| Data Source | Nature | Period (Precipitation) | Period (Runoff) |
|---|---|---|---|
| TRMM 3B43 | Remote Sensing Observation | 1998-2016 | - |
| ABH Loukkos (Loukkos) | River gauge stations | 1961-2017 | 1971-2017 |
| ABH Loukkos (Oued Laou) | River gauge stations | 1964-2017 | 1971-2012 |
| ABH Sebou (Mjara) | River gauge stations | 1978-2017 | 1959-1996 |

**Table 2.** Datasets used for the study.

### 3.1.2 Gauge station data

Gauge station data was also used. The sources of data that have been used here are: river basin agency of Sebou and river basin agency of Loukkos. In Morocco, river basin agencies (ABH *"Agence du bassin hydraulique"*) are in charge of the evaluation, the planning and the general management of water resource within a watershed. In our case, Mjara watershed is managed by ABH Sebou (contoured with black in Figure 4). For this watershed, 17 rainfall stations and 24 runoff stations are available. Loukkos and Oued Laou on the other hand, are managed by ABH Loukkos. This agency made 11 rainfall and runoff stations available for the purpose of this study. All the gauge stations used in this study are displayed in Figure 5.

These data has the advantage of being more accurate than the spatial data, but are more likely to include missing values. In



| Model name | ATM resolution (lat x lon) | OCN resolution (lat x lon) | Main reference |
|---|---|---|---|
| ACCESS 1.0 | $1.25° \times 1.875°$ | $1/3° \sim 1°$ | [Bi et al. (2013)] |
| CCSM4 | $0.9° \times 1.25°$ | $1.11° \times 0.27° \sim 0.54°$ | [Gent et al. (2011)] |
| GFDL-ESM2M | $2° \times 2.5°$ | $1° \times 1/3° \sim 1°$ | [Dunne et al. (2012)] |
| IPSL-CM5A-LR | $1.875° \times 3.75°$ | $2° \times 0.5° \sim 2°$ | [Dufresne et al. (2013)] |
| MPI-ESM-MR | T63 ($\sim 1.875°$) | $0.4° \times 0.4°$ | [Giorgetta et al. (2013)] |

**Table 3.** Description of global climate models selected as lateral boundary forcings for MRCM simulation in the study.

order to use them, one method of filling the missing values ($\sim 5\% - 10\%$) is to replace them by their longterm mean. There are other methods that can be used to fill the missing value, but only this method will be used in this study, since it seemed accurate enough for our purposes.

### 3.2 Regional Climate modeling

The present study used the MIT-Regional Climate Model (MRCM) based on ICTP-Regional Climate Model Version 3 (RegCM3, Pal et al. (2007)) but with several improvement. A detailed description of the MRCM is given by Im et al. (2014a). There are substantial differences between MRCM and RegCM3. In particular, based on the simulations using MRCM-IBIS over the West Africa, the use of IBIS as the land surface scheme results in better representation of surface energy and water budgets in comparison to RegCM3-BATS. Furthermore, the addition of a new irrigation scheme to IBIS makes it possible to investigate

the effects of irrigation over West Africa (Im et al. (2014b); Alter et al. (2015)).

Building on the good performances of MRCM in previous studies, we project multi-model ensemble regional climate change using this model to advance our understanding of the future conditions in response to anthropogenic GHGs (greenhouse gases) over Morocco. To achieve this, a total of fifteen 31-year projections under multi-GCMs (Global Climate Model: IPSL-CM5A-LR, GFDL-ESM2M, CCSM4, MPI-ESM-MR, ACCESS1.0, Table 3) and multi-scenarios of emissions (Historical, RCP4.5,

RCP8.5) are dynamically downscaled using MRCM with $12 \; km$ horizontal resolution over Morocco (Figure 6). The first year of MRCM simulations in both the baseline and future periods is used for model spin up and has been discarded in the analysis. Five GCMs are selected based on a rigorous evaluation (NRMSE "*Normalized Root Mean Square Error*", PCC " *Pattern Correlation Coefficient*" and annual cycle) of their performances in simulating regional climate over Morocco while RCP4.5 and RCP8.5 represent mitigation and business-as-usual scenarios, respectively. We refer the reader to the companion paper (Kang

et al. (2019)) for a detailed discussion about the RCM simulations and model performance.

### 3.3 Runoff Elasticity

The issue of sensitivity of runoff to climate change has been investigated in different studies. There are several methodologies to address this issue as reported by Sankarasubramanian et al. (2001). One of the most common approaches is to use a calibrated hydrological model. Then, with the help of RCM's simulations of future climate, one is able to study the sensitivity of runoff.





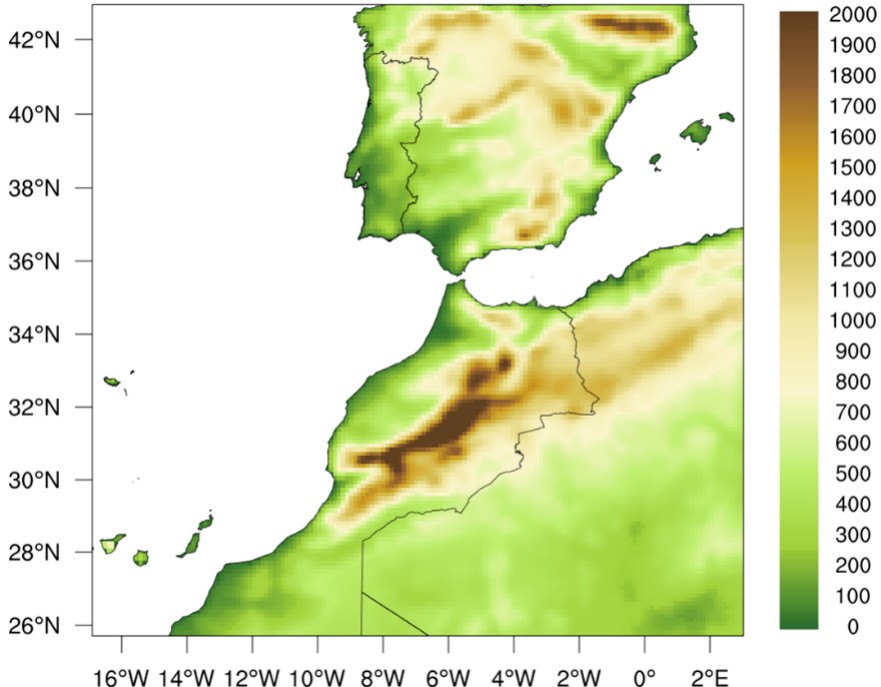

**Figure 6.** MRCM simulation domain and topography (unit: meter).

The major difficulty of this method lies in the calibration of model parameters from a finite set of observations. One assumes that the parameters values do not change through time. This assumption has been extensively discussed in the community, and several studies demonstrated the importance of a dynamical correction of model parameters (Reichle et al. (2002); Moradkhani et al. (2005)) .

5    Another approach is the application of simple water balance models, and to anatically derive elasticity coefficient. One could explain the sensitivity of runoff to all climatic variables as in (Yang and Yang (2011)). Indeed, this study concludes that the hydrological cycle is influenced by other climatic variables than precipitation and temperature, and compute runoff sensitivity to net rediation and wind speed at a height of 2m above the ground. On the other hand, following *Budyko* hypothesis, the long term mean runoff is mainly driven by the long term mean of precipitation and potential evapotranspiration rate. Therefore,

10    runoff elasticity coefficient should be an explicit function of precipitation and potential evapotranspiration rate. Following this idea, different studies have developped empirical relationships (Schreiber (1904); Ol'Dekop (1911); Turc (1953); Pike (1964); Zhang et al. (2001)). Niemann and Eltahir (2005) demonstrated that the estimation of runoff elasticity coefficient with these analytical formulas is consistent with the one calculated from a physically based hydrological model (Niemann and Eltahir (2004)).

15    Another method to derive runoff sensitivity to climate change is the use of observations datasets. One is able to examine how runoff changes as climatic variables change. This assumes that the response of annual runoff to the variability of annual





| Fonctionnal form | Name | Source |
|---|---|---|
| $F'(\phi) = e^{-\phi}$ | Schreiber | Schreiber (1904) |
| $F'(\phi) = tanh(\frac{1}{\phi}) - \frac{4}{\phi(e^{-\frac{1}{\phi}}+e^{\frac{1}{\phi}})^2}$ | Ol'dekoop | Ol'Dekop (1911) |
| $F'(\phi) = -\frac{1}{\phi^3 \left[1+(\frac{1}{\phi})^2\right]^{3/2}}$ | Turc-Pike | Turc (1953)-Pike (1964) |
| $F'(\phi) = \frac{\frac{2}{\phi}+\frac{1}{\phi^2}}{(1+\phi+\frac{1}{\phi})^2}$ | Zhang | Zhang (2009) |

**Table 4.** Mathematical form of $F'(\phi)$.

climatic forcing is the same as the response of the long term mean runoff to the changes of the long term mean climatic variables. This approach has been tested in Risbey and Entekhabi (1996); Sankarasubramanian et al. (2001). The first study showed that Sacremento river runoff is insensitive to changes in temperature while it is more sensitive to precipitation. In the second study, the authors performed a similar methodology in the major rivers of the United States. They showed that when

the rivers are located in dry areas, the sensitivity of runoff is mostly explained by precipitation. However when the rivers are located in areas with an important contribution of snow-packs to water balance, the runoff is less sensitive to precipitation.

In the present study, as we lack sufficient data to develop a well calibrated hydrological model, we choose to evaluate the potential change in runoff with the last two approaches described above. First, we evaluate the potentiel change in runoff through empirical equations first defined by (Dooge et al. (1999); Arora (2002)). Following these studies, long term change

in runoff is mainly driven by the long term change in precipitation and the long term change in potential evapo-transpiration (PET). Therefore, one can write:

$$\frac{\Delta Q}{Q} = \epsilon \frac{\Delta P}{P} - \epsilon_{PET} \frac{\Delta E_0}{E_0}. \tag{1}$$

Where $\frac{\Delta Q}{Q}$, $\frac{\Delta P}{P}$ and $\frac{\Delta E_0}{E_0}$ are long term normalized changes in runoff, precipitation and potentiel evapotranspiration respectively. Consequently, the evaluation of runoff sensitivity to climate change requires the estimation of $\epsilon$ (runoff sensitivity to

precipitation)and $\epsilon_{PET}$ (runoff sensitivity to PET). In Arora (2002); Yang and Yang (2011) these coefficients are computed using empirical equations:

$$\epsilon = 1 + \beta. \quad \epsilon_{PET} = \beta \tag{2}$$

where $\beta = \frac{\phi F'(\phi)}{1-F(\phi)}$. The estimation of the runoff sensitivity coefficients are therefore a function of the aridity index $\phi$ and the mathematical function $F(\phi)$ relating the runoff ratio to precipitation (or PET) to the aridity index. These fonctions have been

defined as empirical relations in different studies. Although those formulas have been developed to find the ratio of evapo-transpiration rate over the potential evapotranspiration rate, they could be rearranged to runoff sensitivity to climate change as in Niemann and Eltahir (2005); Yang and Yang (2011)(See Table 4). In the present work, we compare the results from different empirical relationships. The long term change in precipitation is estimated through RCMs (Regional climate model simulations), and the long term change of potential evapotranspiration is derived from Penmann-Monteith equation (Monteith





et al. (1965)) and using RCMs simulations results. This method is preferred rather than Thornthwaite (Thornthwaite and Mather (1957)) equation for the reasons described in Yang et al. (2017).

The second approach used here is to evaluate the changes of runoff as a result to climate change through data analysis. The

sensitivity of runoff to climate change is computed through the runoff elasticity coefficient. First we use the definition given by Schaake et al. (1990). The Runoff Elasticity ($\epsilon(Q)$) is defined as the fractional change in runoff ($Q$) due to a fractional change in precipitation ($P$) as shown in the following equation:

$$\epsilon(Q) = \frac{\partial Q}{\partial P} \cdot \frac{\overline{P}}{\overline{Q}} \tag{3}$$

In the present case the estimation is done with the following parameters: $\overline{P}$ is the annual mean precipitation, $\overline{Q}$ is the annual

mean runoff and $\frac{\partial Q}{\partial P}$ is estimated using linear regression.

## 4 Results

### 4.1 Hydrology of different watersheds

#### 4.1.1 Loukkos

Loukkos watershed is one of the wettest regions in Morocco. The annual precipitation there is around $790\ mm$ and the hydrological regime is characterized by a wet winter and a dry summer with a maximum of precipitation during November and/or December (see Figure 7(a)). However, precipitation there varies significantly from year to year. The annual precipitation could reach $1400\ mm$ in some years (See Figure 7(b)). Indeed, the coefficient of variation of annual precipitation computed using the ABH Loukkos data is $28\%$. Given this variability, water resources management is a challenging task.

The strong inter-annual variability of precipitation leads to a strong inter-annual variability in runoff (see Figure 7 c)). The pattern of runoff is similar to those of precipitation. The annual mean runoff is $1\ km^3/yr$. Given its topography, this watershed represents a good site for Morocco to store water. However, this region lacks the infrastructure needed to store water and to optimize water resources management.

#### 4.1.2 Oued Laou

The watershed of Oued Laou is the smallest one considered in this analysis. However due to its geographical position, the region receives a large amount of precipitation compared to the other areas in the country. The annual mean precipitation is about $740\ mm$ which represents a large quantity of water falling on the relatively small area of the watershed ($919\ km^2$). However, the variability of precipitation is tremendous as the coefficient of variation is $43\%$ (see Figure 8), which makes the



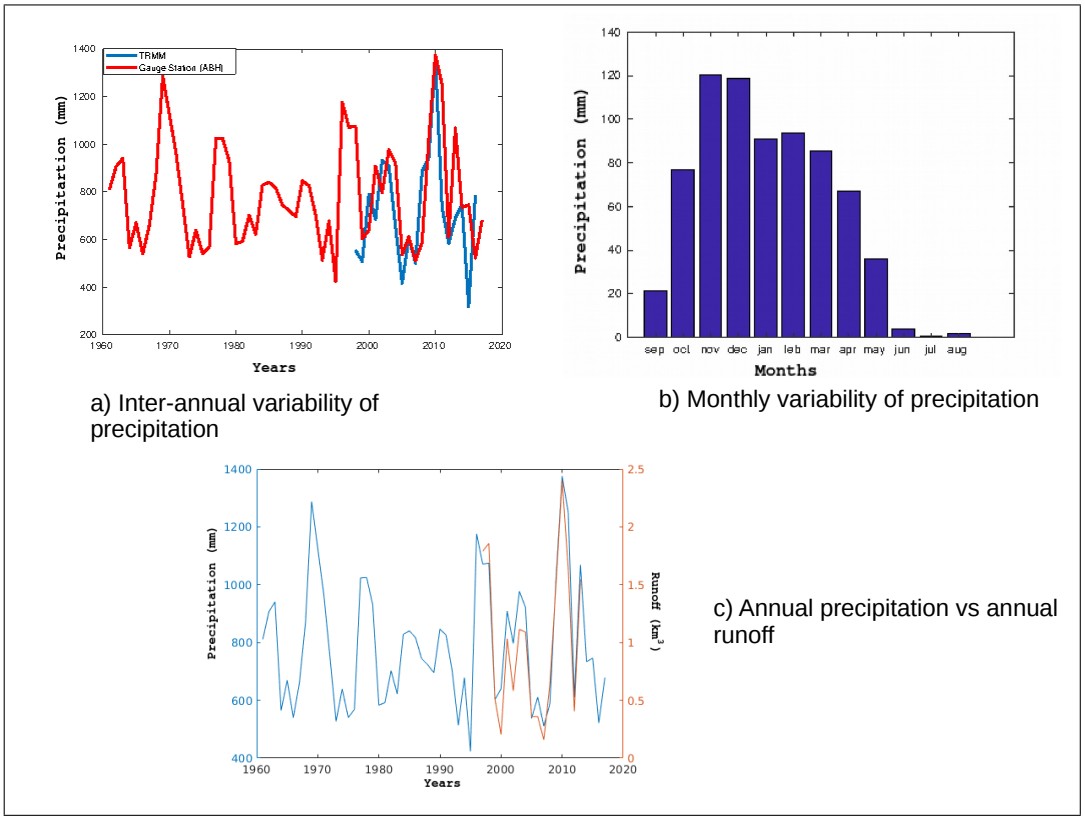

a) Inter-annual variability of precipitation

b) Monthly variability of precipitation

c) Annual precipitation vs annual runoff

**Figure 7.** Hydrology of Loukkos watershed

forecasting of water availability very difficult. The plot in Figure 8 c) presents the variability of the runoff and precipitation over Oued Laou. The variability of runoff seems to follow that of precipitation. The annual mean runoff in Oued Laou is $0.4 \ km^3$.

### 4.1.3 Mjara

The hydrology in Mjara is also characterized by a significant monthly variability. The annual mean precipitation is $655 \ mm$, and the coefficient of variation is about $43\%$. Thanks to gauge station at Mjara, we were able to estimate runoff. As seen in Figure 9 c), the variability of water availability at Mjara follows the variability of precipitation. This statement is especially remarkable during the drought years, that Morocco has experienced during the $1990'$, when the variability of runoff has followed very tightly that of precipitation. All in all, the annual average of runoff from Mjara, considered as the wettest region in Morocco and where the largest (namely *AlWahda*) dam of the country is located, is $3 \ km^3/yr$.

### 4.2 Future changes in precipitation

In order to assess climate change impact on water availability, we have performed numerical simulations (Kang et al. (2019)) as described in section 3.2. Overall, the resulting simulations forced by the 5 GCMs agree that precipitation will decrease in the

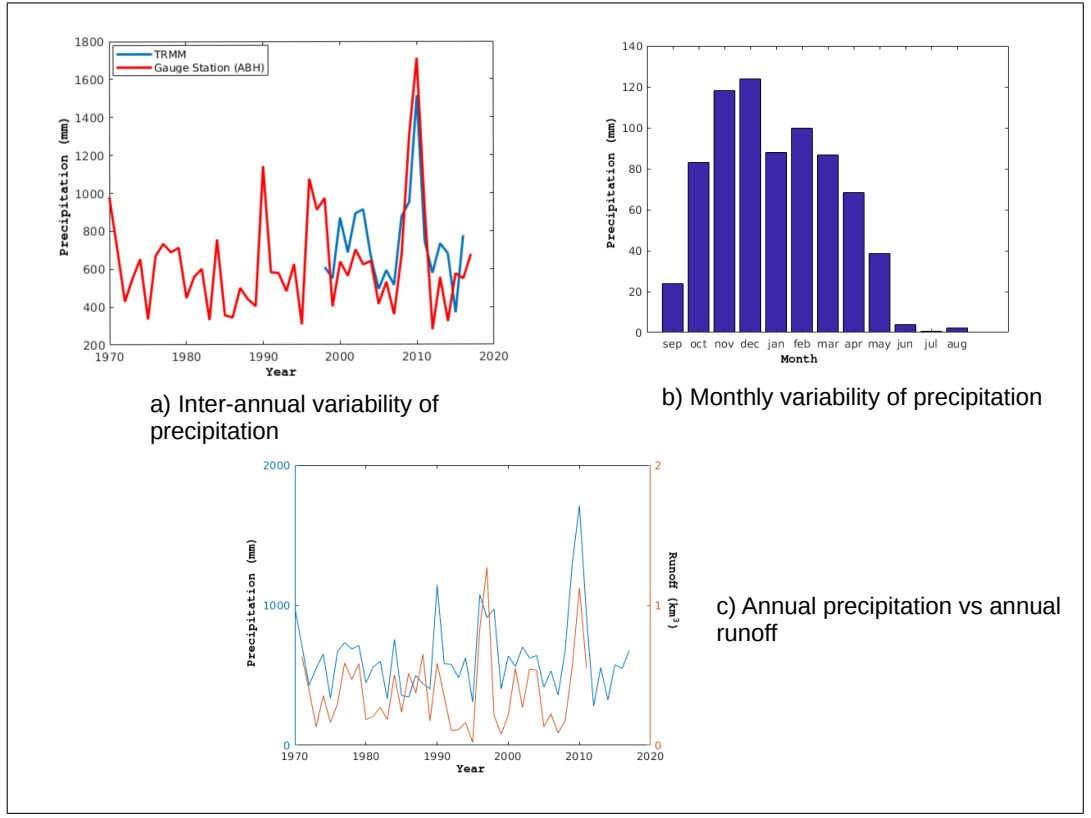

**Figure 8.** Hydrology of Oued Laou watershed

future for the region of interest. However depending on the boundary condition and on the scenario considered, the magnitude of the decrease is different. All of these results are summarized in Table 5. There is a large uncertainty about the amount of precipitation in the future for the study region. As it was shown in earlier studies (Déqué et al. (2007)) the greater source of uncertainties in RCMs simulation remains to be boundary condition forcing. This would explain the difference in results in our case when we change the forcing GCMs, even for the same scenario. However, for one fixed boundary conditions and for the same scenario, the amount of precipitation decrease is similar over the three watersheds. Following RCP $4.5$, the minimum of the decrease is simulated when the forcing GCMs is CCSM, the estimated amount of change is roughly around $-10\%$. The maximum of the decrease is simulated when the model is forced with ACCESS, and the amount is roughly equal to $-35\%$. As a result, following RCP $4.5$, the projected quantity of precipitation change is between $-35\%$ and $-10\%$ for this region. On the other hand following RCP $8.5$, the minimum of the decrease is achieved when the boundary conditions are described by MPI and is roughly equal to $-33\%$, however the maximum of the decrease is simulated when the RCM is forced by IPSL, and is estimated to be around $-70\%$. As a result, if the future is driven by the scenario RCP $8.5$, the precipitation are expected to decrease by $33\%$ in the best case and by $70\%$ in the worst case. On average, following RCP $4.5$ precipitation is going to decrease by $24\%$, while under the scenario RCP $8.5$ the decrease would achieve $47\%$. To sum up, precipitation is likely going

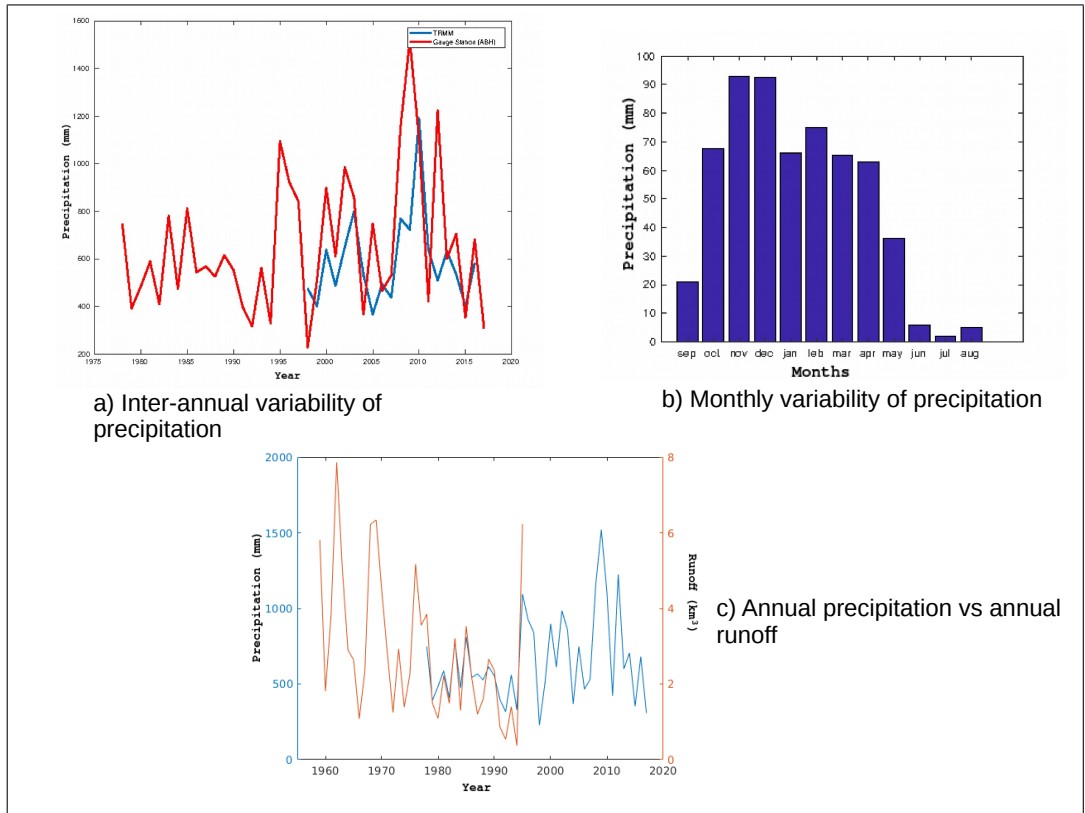

a) Inter-annual variability of precipitation

b) Monthly variability of precipitation

c) Annual precipitation vs annual runoff

**Figure 9.** Hydrology of Mjara watershed

to decrease in the future for this region, however the magnitude of the decrease is uncertain. The range of uncertainty has been quantified.

### 4.3 Future changes in Runoff

#### 4.3.1 Using data analysis

5  The first approach used here to assess climate change impact on water availability is the estimation of runoff elasticity coefficient through analysis of available data. We follow the definition described in section 3.3. The objective is to estimate the potential loss of water that would result from the decrease of precipitation using available precipitation and runoff data over each watershed. We assume that the relationship between precipitation and runoff is well represented by past data. This relationship is first described by the slope of the runoff-precipitation relationship as documented in observations (see Figure 10).

10  Then we normalize this relationship by dividing with the ratio of long-term runoff to long-term precipitation.

For the specific case of Oued Laou, only one precipitation gauge station is present. In order to alleviate bias creation in the calculation of the elasticity coefficient, we use TRMM observations. Figure 10 shows the estimation of runoff elasticity coeffi-



|  |  | Oued Laou | Mjara | Loukkos |
|---|---|---|---|---|
| MPI | RCP4.5 | -16% | -17 % | -19% |
| MPI | RCP 8.5 | -32 % | -34% | -33% |
| GFDL | RCP4.5 | -27% | -28 % | -27% |
| GFDL | RCP 8.5 | -40 % | -44% | -43% |
| IPSL | RCP4.5 | -36% | -31 % | -36% |
| IPSL | RCP 8.5 | -69 % | -71% | -69% |
| CSM | RCP4.5 | -16% | -0.5 % | -11% |
| CSM | RCP 8.5 | -36 % | -37% | -36% |
| ACCESS | RCP 4.5 | -34% | -36 % | -36% |
| ACCESS | RCP 8.5 | -50 % | -54% | -52% |

**Table 5.** A summary of the potential changes in precipitation following the $RCP4.5$ and $RCP8.5$ simulations with MRCM for the 5 different GCMs forcing.

|  | Oued Laou | Mjara | Loukkos |
|---|---|---|---|
| Elasticity | 1.65 | 1.6 | 1.7 |
| Turc Pike | 1.8 | 1.6 | 1.6 |
| Ol'Dekoop | 1.6 | 1.4 | 1.4 |
| Zhang | 1.9 | 1.7 | 1.7 |
| Schreiber | 1.8 | 1.6 | 1.6 |

**Table 6.** Elasticity of runoff to precipitation estimation comparison with analytical formulas

cient for the study region. The three watersheds seem to behave with the same manner when forced by precipitation variability. In general, the elasticity coefficient is around 1.6 which means that a change of 10% in precipitation would lead to 16% change in runoff.

Based on the calculation of runoff elasticity coefficients and given the projected decrease of precipitation described in the previous section, we are able to estimate the potential total available water in the three watersheds in the future. Table 7 summarizes the results. As a metric of the vulnerability of water availability in the region to the project, we show the ratio of the total amount of water to be transferred ($0.86$ $km^3/yr$) to the total available water. In the current climate, about 20% of the total available water is allocated for the the *Water Highway* in the initial design. On average, following the projection RCP 4.5 it is expected that only 40% of the actual available water would be lost in the future compared to the current climate. As a result 33% of the available water will be allocated for the project if the design is kept the same as in the current climate. Consequently, the project will probably turn in the region into a water-scarcity one. Given this analysis and to ensure that the project does not impact the region negatively, the amount of the transfer should be rescaled down. On the other hand, following





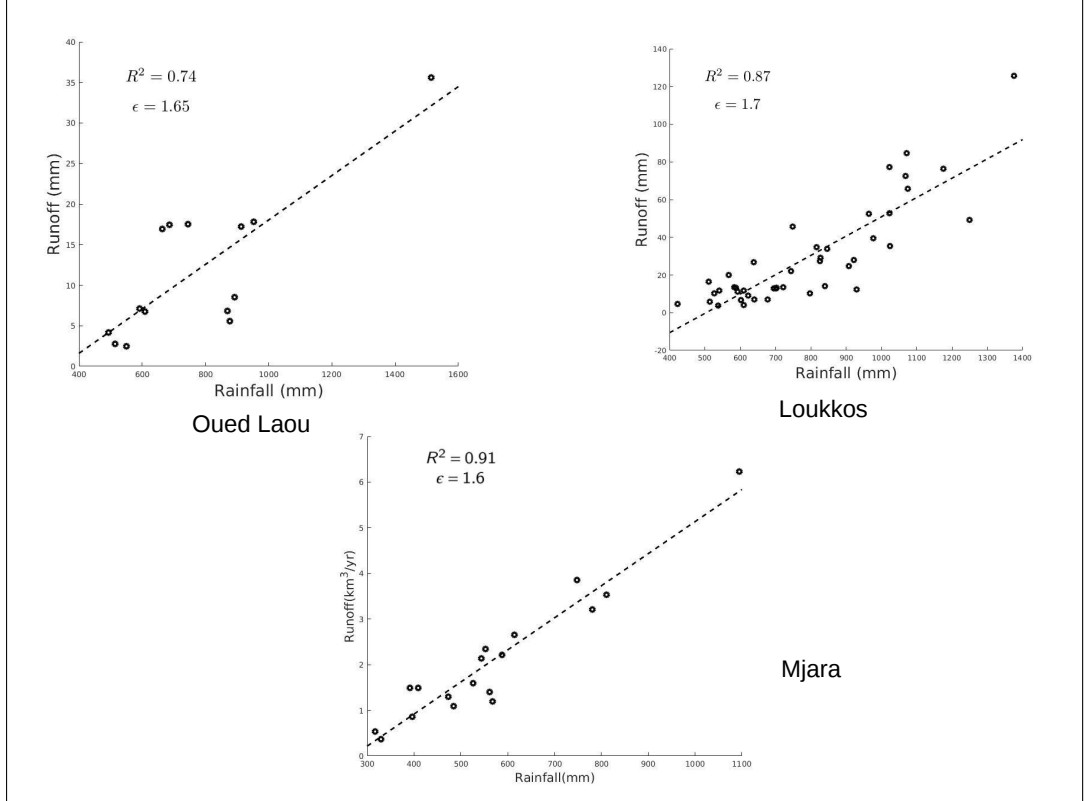

**Figure 10.** Elasticity coefficient estimation for the three watersheds with regression method

RCP $8.5$ only $27\%$ of actual quantity of water is expected to be left in the region, meaning that the total available water in the region would be $1.19 km^3/yr$. If the project is kept as in the initial design, $72\%(=\frac{0.86}{1.19})$ of the total available water in the region will be transferred. Given this result, the project becomes infeasible, as the local precipitation would probably be short of satisfying the local water demand.

### 4.3.2 Using empirical formulas and RCMs

As described in section 3.3, we consider here the effect of climate change on runoff. First $\epsilon$ and $\epsilon_{PET}$ are estimated using empirical formulas according to Table 4. Table 8 reports the values of $\epsilon$ and the comparison with the corresponding ones from regression analysis, we remind that $\epsilon_{PET} = 1 - \epsilon$. These results are similar to previous conclusions on sensitivity changes in runoff in arid and semi-arid areas Sankarasubramanian and Vogel (2003). Moreover as we compare the results of data analysis with the empirical equations (see Table 6), the results are similar. The sensitivity of runoff to precipitation is around $1.6$. In the work of Tang and Lettenmaier (2012), the authors estimates runoff sensitivity to changes in precipitation for the major global rivers in the world. We found a good agreement between their results for this particular region and ours. According





| | | Oued Laou $(km^3/yr)$ | Mjara $(km^3/yr)$ | Loukkos $(km^3/yr)$ | Available water $(km^3/yr)$ | $\frac{\text{Project}}{\text{Total}}(\%)$ |
|---|---|---|---|---|---|---|
| | Present | 0.4 | 3 | 1 | 4.4 | 20 % |
| MPI | RCP45 | 0.3 | 2.2 | 0.68 | 3.18 | 27% |
| MPI | RCP85 | 0.19 | 1.4 | 0.44 | 2.33 | 42% |
| GFDL | RCP45 | 0.22 | 1.4 | 0.5 | 2.12 | 40% |
| GFDL | RCP85 | 0.14 | 0.81 | 0.27 | 1.22 | 70% |
| IPSL | RCP45 | 0.16 | 1.47 | 0.39 | 2.02 | 43% |
| IPSL | RCP85 | 0 | 0 | 0 | 0 | - |
| CSM | RCP45 | 0.3 | 2.76 | 0.82 | 3.88 | 22% |
| CSM | RCP85 | 0.16 | 1.23 | 0.39 | 1.78 | 48% |
| ACCESS | RCP45 | 0.17 | 1.26 | 0.39 | 1.82 | 47% |
| ACCESS | RCP85 | 0.07 | 0.42 | 0.12 | 0.62 | 139% |

**Table 7.** A summary of the expected loss of water in the three watersheds, and how the project will affect water availability. We are assuming the project design of transferring $0.86 km^3/yr$. The available water is referring to how much water is available in the whole study region. $\left(\frac{\text{Project}}{\text{Total}}\right)$ is the ratio of the amount of water to be allocated for the project, if kept as in the original design, over the potential total available water in the study region following each simulation.

| | Oued Laou | | Mjara | | Loukkos | |
|---|---|---|---|---|---|---|
| | RCP4.5 | RCP8.5 | RCP4.5 | RCP8.5 | RCP4.5 | RCP8.5 |
| $\beta$ | 0.8 | | 0.6 | | 0.6 | |
| $\frac{\Delta P}{P}$ | $-25\%$ | $-45\%$ | $-22\%$ | $-48\%$ | $-26\%$ | $-47\%$ |
| $\frac{\Delta E_0}{E_0}$ | 11% | 22% | 11% | 25% | 10% | 21% |
| $\epsilon\frac{\Delta P}{P}$ | $-45\%$ | $-81\%$ | $-35\%$ | $-77\%$ | $-42\%$ | $-75\%$ |
| $\epsilon_{PET}\frac{\Delta E_0}{E_0}$ | 9% | 18% | 7% | 11% | 6% | 13% |
| $\frac{\Delta Q}{Q}$ | $-54\%$ | $-99\%$ | $-42\%$ | $-92\%$ | $-48\%$ | $-88\%$ |
| Future available water $(km^3/yr)$ | 0.18 | 0.04 | 1.74 | 0.24 | 0.52 | 0.12 |

**Table 8.** Summary of runoff sensitivity to precipitation and PET under climate change and the estimation of future total available water in the region

to the results reported in Table 8, change in runoff is largely sensitive to precipitation. If we compute the sensitivity index of precipitation changes $S_P$ according to the index defined by Saltelli (2002), we find $S_P = \frac{\epsilon^2}{\epsilon^2 + \epsilon_{PET}^2} = 0.88$. Meaning that $88\%$ of the changes in runoff due to climate change are attributed to changes in precipitation. Ultimately changes in PET contribute only to $12\%$ changes of runoff.

5 Following Eq.(1) and in order to fully evaluate changes of runoff due to climate change, an assessment of changes of PET is needed, as the changes of precipitation has already been discussed in previous section. Using Penmann-Monteith equation





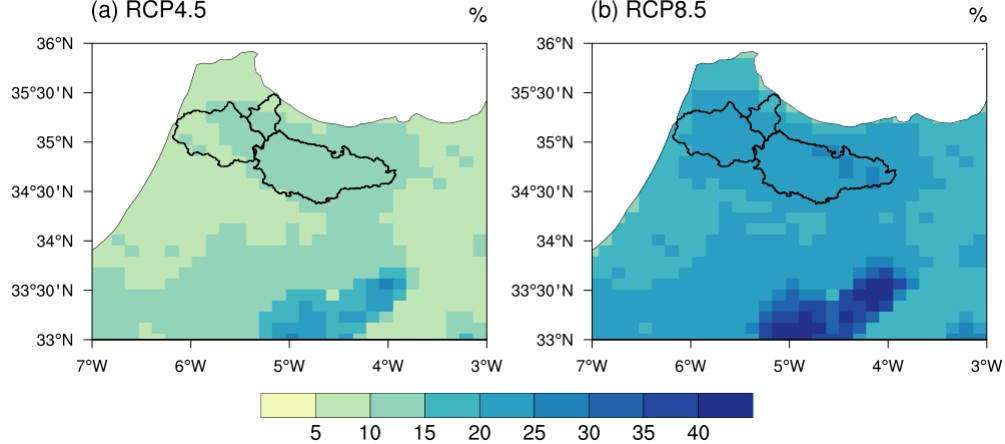

**Figure 11.** Future changes in PET computed using Penmann-Monteith equation and RCMs following RCP 4.5 (a) and RCP 8.5 (b) respectively. The three watersheds of interest are contoured

and RCMs results, we plot figure 11, where we show changes in PET according to RCP 4.5 and RCP 8.5 respectively. The watersheds of interest are contoured in black in the map. Following RCP 4.5, PET is going to increase by 11% on average, whereas the increase will achieve 22% following RCP 8.5. All those results are summarized in Table 8.

Based on this analysis, we are able to say that future water availability is very likely to decrease in the region. This decrease is

mainly explained by the two factors that are described in Eq.(2):

1. Precipitation which is the main water supplier in the region will likely decrease significantly.

2. The increase of PET will lead to increase in the loss of available water by evapotranspiration due to the presence of significant energy at the surface, described by the increase of temperature in the simulations.

As a result we are able to quantify the future potential changes of available water in the region as reported in Table 8. Following

RCP 4.5, the study area will likely lose 44% of its actual available water, meaning that the region will have only 2.44 $km^3/yr$ for supply. Therefore and given the actual scale of the project 44% of the total available water will be transferred. Following RCP 8.5, the region is very likely to suffer water scarcity, as the supply reduce dramatically.

In this study, due to lack of data we were not able to perform hydrological modeling. In their report, RICCAR [Report (2017)] were able to assess the vulnerability of water resources in the whole arab region to climate change. As a matter of fact, they

used RCMs results to force two hydrological models (VIC and HYPE) to assess changes in runoff over all the Arab region. Their conclusion for Moroccan highlands, where the sensitivity of runoff to other parameters like vegetation have been token into account, are similar with the findings of our research: All the coupled climate-hydrological models simulations agree that the region is a subject to a decrease in runoff due to a decrease in precipitation and an increase in temperature. Moreover, under RCP 4.5, the decrease of runoff in the region ranges in $[-40\%, -20\%]$, and in $[-60\%, -50\%]$ under RCP 8.5. The main

differences with our study come from the fact that the hydrological modelling was performed over a global scale.





### 4.3.3 Feasability of the *Water Highway* project

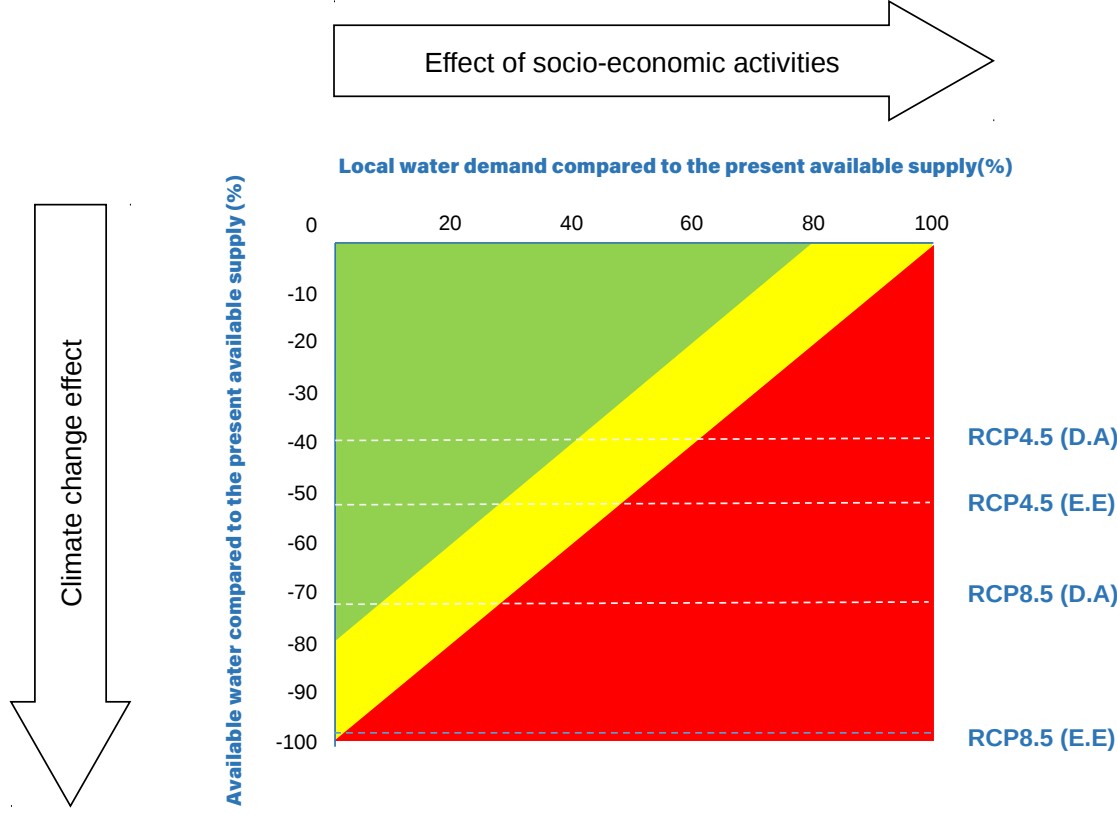

**Figure 12.** Mapping of vulnerability function of local water demand and supply. The green region is where the project is feasible, the yellow where it should be rescaled and the red where it cannot be feasible. D.A: Data analysis, E.E: Empirical equations.

Based on the previous analysis, we are able to address the problem of viability of the *Water Highway* project. The project aims to transfer $0.86\ km^3/yr$ from the northern watersheds considered in this analysis (Oued Laou, Loukkos and Mjara) to the southern regions. The numerical simulations have proven that the donor watersheds are going to lose significant amount of precipitation due to global warming, therefore water availability would also decrease. Following RCP 4.5, the results based on estimation of runoff elasticity from data analysis shows that the expected (average from the different simulations) amount of loss is $40\%$, while the second study based on the empirical equations shows that the decrease could reach $44\%$. On the other hand and following RCP 8.5 the first analysis forecast $73\%$ of loss while the second shows that the region would suffer from severe water scarcity with little water available in rivers.





To make things worse, climate change is not the only limiting factor in the region. The northern part of Morocco has known
lately an intensification of water demanding activities, like agriculture. Therefore, the feasibility of the project depends also
on the amount of water allocated to satisfy the local demand. Figure 12 maps the regions where the project could be feasible
(green), need to be rescaled (yellow) or not feasible at all (red), according to an assumed level of water demand (computed

on the basis of the present available water) and the supply given by each scenario and each method used to estimate water
availability. While the vertical axis corresponds to the stress caused by climate change, the horizontal axis shows how the
socio-economic activities will affect the feasibility of the project. The horizontal line of the figure, corresponding to zero
in the vertical axis, represents the present state. Under current climate the design of *Water Highway* requires allocation of
$20\%$ of the total available water ($\frac{0.86}{4.4}$). That leaves $80\%$ of the available water to satisfy local water demand. We plot also

lines corresponding to the projected water quantities that are going to be lost following RCP $4.5$ and RCP $8.5$ respectively.
Following the mitigation projection, the northern regions are going to lose $40\%$ (based on data analysis study) and $44\%$ (based
on the second analysis) of the actual water quantities. This means that if the project is kept as in the initial design of the current
available water of $4.4\ km^3/yr$, the water demand in the region should not exceed $36\%$ ($20\% + 44\% + 36\% = 100\%$) of the
present available water. On the other hand, if the future is driven by business-as-usual projection, the project is unlikely to be

feasible as the supply would definitely not be sufficient to satisfy local demand. Little water is likely to flow in streams ($99\%$
of the total actual available water will likely be lost).

## 5 Conclusions

The North-South Water Transfer project (*Water Highway*) in Morocco aims to supply the vulnerable regions in the south
with water. The project will benefit from the excess water in the northern regions to transfer $0.86\ km^3$ each year. This water

transfer will follow three phases, and it will include the building of two dams, several pumping stations and $500\ km$ of pipes
from the northern region southward until Al Massira dam. The project will contribute to alleviate water stress in the south,
especially supporting the sustainability of agriculture. However despite of its great potential positive impacts, the project is
largely sensitive to the amount of water that will be available in the future. Therefore, its feasibility remains unclear given
climate change, which is going to affect precipitation over Morocco in general. Regional climate models (RCMs) agree in

predicting that the future will bring less water. However, there is a significant uncertainty about the quantity of water that
will be lost. There are several reasons behind this uncertainty in the simulations results. Boundary conditions given by GCMs
(Global circulation models) are the main contributors to this uncertainty. Furthermore, the future simulations are also driven by
different scenarios of greenhouse gas emissions, which lead to different climate projections. In the present study, we have used
two projections: RCP $4.5$ (with mitigation) and RCP $8.5$ (business as usual). Following each of the aforementioned scenarios,

the precipitation will decrease in the northern water basins. Following RCP $4.5$ the amount of the decrease is expected to be
in the range $[-33\%, -10\%]$. In comparison, following RCP $8.5$ the amount of the decrease is expected to be in the range
$[-70\%, -34\%]$.
In order to assess the water quantities that will be lost, we used two different approaches: data analysis and sensitivity analysis





using empirical equations. In the first approach, we have computed the runoff elasticity coefficients based on field observations. Overall this analysis shows that a 10% change in precipitation will lead to 16% change in runoff. Compared to the results of the expected decrease in precipitation, the elasticity analysis indicates that runoff is going to be even more sensitive to climate change. In the second approach, we assessed the sensitivity of runoff to climate change through different empirical

relationships. The conclusions are similar to previous results. We found that changes in runoff is mainly driven by the decrease of precipitation. Furthermore, we estimated the potential amount of loss: following RCP 4.5 runoff is going to decrease by 44%, however under RCP 8.5 a very little amount of water is likely to flow in rivers as 99% of the total available water is likely to be lost.

Finally, we have developed a map of vulnerability given each climate simulation and the future local water demand. This

analysis have helped to assess the viability of the project based on assumed water demand in the donor watersheds. Under RCP 8.5 the project is likely unfeasible. However under RCP 4.5 a rescaled version of this project may be feasible depending on how much is assumed for the local water demand. Our results agree in general with the conclusions of Greve et al. (2018) who discussed the issue with the global scale. Given all the uncertainties introduced by climate change and reported in this study, water transfer project as an example of water resources planning needs to be carried carefully.

Based on this work, there are several questions that still need to be addressed. First, given the large uncertainties in the climate model simulation results and the climate scenarios, further research is needed to assess this uncertainty. Second, given the complex hydrological behaviour of the watersheds, there is a need to perform hydrological modelling in order to assess more accurately how climate change will affect the runoff. Finally, in order to address a global critical analysis of the feasibility of the project, the future local water demand driven by the expected population growth and the future water demanding activities

should be quantified.

*Acknowledgements.* This work has been done under the UMRP Project with a financial support from OCP. We are grateful to Alexandre Tuel, Catherine Nikiel, Timothy Adams and the rest of Eltahir Research Group for their helpful comments. The authors would like to thank Abdellah Bourak from ABH Sebou for his help to get data. In the same way, we would like to acknowledge the work of Abdelouahed El Kouri and Rachid Chahri and Salah Eddine Dahbi from ABH Loukkos for all their help and discussion.



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
