# Peer review of "Climate Change impacts on the Water Highway project in Morocco"

_Hydrology and Earth System Sciences, 2019_

## Referee Comment (RC1) · Anonymous Referee #1 · 26 Jul 2019

Although the manuscript of et El Moçayd al. is well written (besides some spelling or grammar errors), I believe it does not fit in its current form in the scope the HESS journal. The analyses are rather simple and more important they do not provide new results. The manuscript reveals several trivial sentences, such as "change in runoff is largely sensitive to precipitation", which is quite expected in a water-limited environment such as in Morocco. As noted line 5, page 22, "the conclusions are similar to previous results".

There is no literature review about similar work in Morocco, while the results presented in the present manuscript can be already found in the following published papers:

Tramblay Y., Jarlan L., Hanich L., Somot S. (2018). Future scenarios of surface water resources availability in North African dams. Water Resources Management 32(4),

1291-1306.

Filahi S., Tramblay Y., Mouhir L., Diaconescu E.P., (2017). Projected changes in temperature and precipitation in Morocco from high-resolution regional climate models. International Journal of Climatology 37(14), 4846-4863.

Marchane A., Tramblay Y., Hanich L., Ruelland D., Jarlan L. (2017). Climate change impacts on surface water resources in the Rheraya catchment (High-Atlas, Morocco). Hydrological Sciences Journal 62(6), 979-995.

Droogers P, Immerzeel WW, Terink W, Hoogeveen J, Bierkens MFP, van Beek LPH, Debele B (2012) Water resources trends in Middle East and North Africa towards 2050. Hydrol Earth Syst Sci 16(9):3101–3114.

Schilling J, Freier KP, Hertig E, Scheffran J. 2012. Climate change, vulnerability and adaptation in North Africa with focus on Morocco. Agric. Ecosyst. Environ. 156: 12–26.

Patricola CM, Cook KH. 2010. Northern African climate at the end of the twenty‐first century: an integrated application of regional and global climate models. Clim. Dyn. 35: 193–212.

Some specific comments:

Section 4.1 is not really a "result" section, it is rather a presentation of the study area and data.

Section 4.3.1, before using TRMM precipitation, a validation against observed data would be welcome.

Moreover, no validation of the RCM simulations is provided with observations.

The figures are not clear, such as figures 7 8 9 10.

Recommendations:

An interesting contribution of this article would be to provide an assessment of changes in runoff sensitivity coefficients (to precipitation and evapotranspiration changes), for both the historical periods and the RCPs. For instance, the coefficients of equation 2 could be computed with each RCM simulation and for different time periods and basins, to evaluate the spatio-temporal patterns and how they evolve in time, taking into account the inter-model spread in RCM simulations. In addition to reference Penman evapotranspiration, they could also analyze the actual evapotranspiration simulated by the RCMs. This requires substantial work and re-submitting a reshaped manuscript.

---

## Author Comment (AC1) · 2 Aug 2019

We would like to thank the reviewer for the feedback about our work. The review suggested several interesting research path. However, we feel that the manuscript has been unfairly evaluated. First the reviewer began by assessing that our work does not fall under the scope of the journal. This remark is deeply contested because the journal clearly specifies in its webpage that it aims to: "encourage and support [. . .] applied research that advances the understanding of hydrological systems, their role in providing water for ecosystems and society [. . .]". The questions addressing the Moroccan case in the present work are consistent with this aim. Furthermore, the review states that the analyses do not provide new results. We think that this statement is not true as the feasibility of the water highway project in Morocco (largest water infrastructure

project in that country) under climate change impacts has not been assessed before in the literature. This manuscript is the unique scientific contribution that provides a deep discussion about how important is this project ? and what are the climate and socioeconomical conditions under which the project would be feasible?. We recall that the aim of this paper is to provide a simple and rigorous methodology for water engineers in Morocco to assess such project. The literature provided by the reviewer addresses the classical problem of the impact of climate change on precipitation and water availability in Morocco. A question to which we answer in our analysis using a higher resolution model MRCM. The validations of the RCM simulations are provided in the companion paper "Kang, S., Tuel, A., & Eltahir, E. A. (2019). High resolution climate change projections over northwest africa. In preparation". Finally, a revision of the present work will include the evaluation of the runoff sensitivity coefficients using the RCMs and all the other minor comments given by the reviewer.

---

## Referee Comment (RC2) · Anonymous Referee #2 · 16 Dec 2019

Overall, this paper presents a simple but useful analysis of the sensitivity of runoff in Morocco to changes in precipitation and PET. The approach is simple due to significant data limitations within the region. While the methods used in the paper are not new, their application to a data-limited environment is new and has potential utility in many other regions. My general comments are as follows: 1. While the simplicity of the approach is warranted in this region, I think the simplicity also necessitates examination of the uncertainties in the approach and results. Where possible the uncertainty should be directly quantified. Where that is not possible, it should be discussed. For example, the authors conclude that the elasticity coefficient is near 1.6 due to the similarity of the three regression slopes in Figure 10. However, each regression slope has significant uncertainty in its value due to the limited number of observations in each plot. The

regression uncertainty is easy to quantify because the authors are using standard regression analysis. It would also be interesting to compare the implied range of elasticity from the historical data to the range inferred from the empirical relationships. Similarly, summarizing the results with single lines in Figure 12 seems to overstate the certainty of the analysis. Could ranges be presented instead? 2. The analysis is largely based on the assumption that dry years in the past produced runoff in the same manner as a dry climate will. An analogous assumption is made for PET changes. I think the paper would be strengthened by a short discussion about factors that may limit the applicability of this approach/assumption (or why this is a good assumption). For example, this approach seems to imply that the within-year variations of rainfall in the dry climate will be similar to the within-year variations of rainfall during dry years. I think it would be helpful to discuss such implicit assumptions (and to evaluate them if possible). It seems like sufficient data are available to investigate the seasonality assumption. It also seems to imply that the vegetation cover changes the same way in response to a climate change as to a single year drought. Can that assumption be justified? Furthermore, the analysis seems to assume independence between the effects of PET and precipitation variables. Such independence would be violated if there is a transition from snowfall to rainfall. Would that confound the analysis? Does the historical record suggest such independence? Is such a transition expected? 3. The analysis makes an implicit steady-state assumption when evaluating the water highway project. Specifically, forecasts of 31 years into the future are used to analyze the project's viability, but those conditions may not be representative of the project design life. Imagine if the design life is 30 years. If built today, the project would experience a range of conditions between the current climate and the predicted climate that might make it more viable. The authors should mention this issue in their paper and provide an argument about why 31 years is appropriate for this assessment. 4. The analysis relies on dynamically downscaled GCM results. However, the paper does not discuss how that downscaling was accomplished and only cites another paper that is in preparation. I find this problematic since the validity of that work has not been substantiated by a peer-reviewed

publication.

Specific Comments 1. The paper needs some additions that help relate the analysis methods to the datasets that are used. Please add explanations about how the variables in the various equations are calculated using the data (e.g., the evaluation of the derivative, the time step of the data, etc.). 2. Some of the variables are not clearly defined, such as phi and F and F'. 3. Can you make a quick analysis to justify why replacing missing data with the mean is appropriate? 4. Table 4. Fonctionnal needs to be translated, I believe. 5. In the first half of the paper, the first sentence or two in several sections is redundant with previous sections. Please remove redundancies.

---

## Author Comment (AC2) · 11 Jan 2020

We would like to thank the reviewer for the positive remarks that will certainly improve the quality of our paper and highlight the originality of the work. We appreciate the quality of the reviewer comments, especially regarding the necessity to address uncertainty, and will respond positively to all of the given remarks. These are our responses to the comments raised in the review:

**Remark 1:** I think the simplicity also necessitates examination of the uncertainties in the approach and results. Where possible the uncertainty should be directly quantified. Where that is not possible, it should be discussed. For example, the authors conclude that the elasticity coefficient is near 1.6 due to the similarity of the three regression slopes in Figure 10. However, each regression slope has significant uncertainty in

its value due to the limited number of observations in each plot. The regression uncertainty is easy to quantify because the authors are using standard regression analysis. It would also be interesting to compare the implied range of elasticity from the historical data to the range inferred from the empirical relationships. Similarly, summarizing the results with single lines in Figure 12 seems to overstate the certainty of the analysis. Could ranges be presented instead?

**Answer:** Uncertainty quantification is surely a relevant question that should be addressed, especially in the context of climate change impact on the conception of hydraulic structures such as the Water Highway. As suggested by the reviewer, uncertainty will be correctly addressed in the revised manuscript. Regression uncertainty will be quantified. Figure 12 will be adjusted according to this analysis.

**Remark 2:** The analysis is largely based on the assumption that dry years in the past produced runoff in the same manner as a dry climate will. An analogous assumption is made for PET changes. I think the paper would be strengthened by a short discussion about factors that may limit the applicability of this approach/assumption (or why this is a good assumption). For example, this approach seems to imply that the within-year variations of rainfall in the dry climate will be similar to the within-year variations of rainfall during dry years. I think it would be helpful to discuss such implicit assumptions (and to evaluate them if possible). It seems like sufficient data are available to investigate the seasonality assumption. It also seems to imply that the vegetation cover changes the same way in response to a climate change as to a single year drought. Can that assumption be justified? Further- more, the analysis seems to assume independence between the effects of PET and precipitation variables. Such independence would be violated if there is a transition from snowfall to rainfall. Would that confound the analysis? Does the historical record suggest such independence? Is such a transition expected?

**Answer:** Every model is built following some unavoidable assumptions. Given the availability of data we had, we were constrained to use the method presented in the paper. However, the remark remains relevant, and an analysis of the seasonality of

precipitation will be added to the manuscript together with some discussion about the assumptions. More discussion about the potential impact and limitations of our methodology regarding the results will be added in the corrected version of our Manuscript. As far as a transition from snowfall to rainfall assumptions is concerned, the studied area receives barely some snowfall such that its role can be neglected in the hydrological cycle.

**Remark 3:** The analysis makes an implicit steady-state assumption when evaluating the water highway project. Specifically, forecasts of 31 years into the future are used to analyse the project's viability, but those conditions may not be representative of the project design life. Imagine if the design life is 30 years. If built today, the project would experience a range of conditions between the current climate and the predicted climate that might make it more viable. The authors should mention this issue in their paper and provide an argument about why 31 years is appropriate for this assessment.

**Answer:** There is mainly two reasons why the 31 years is appropriate for this assessment. The first one is linked to climate change. In fact, this project has been designed in order to face future climate change impacts on the southern Moroccan areas. If the project may jeopardize the water security of the northern region how can it face the southern region water demand as well? The second reason is purely from a feasibility perspective. The project will need more than 10 years to be fully built, not to mention that the project has not started yet (The construction was planned to start in 2018). In addition, the Moroccan government has only made available the budget for the first phase. The second and the third phase are still waiting for the necessary funds, which will certainly delay the project for another 10 years. This discussion will be added in the manuscript.

**Remark 4:** The analysis relies on dynamically downscaled GCM results. However, the paper does not discuss how that downscaling was accomplished and only cites another paper that is in preparation. I find this problematic since the validity of that work has not been substantiated by a peer-reviewed publication.

**Answer:** A manuscript of the companion paper will be included as supplementary

information.

**Specific Comments:** All the specific comments will be addressed in the Manuscript.

---

## Author Response (AR1)

We would like to thank the editor for the positive comments that has surely improved the quality of our paper. All the modifications added to the manuscript are highlighted in blue. We changed all the edits given by the editor and responded positively to all the comments. Moreover, we have also addressed all the comments given by the reviewer and here is a point to point answers to all the comments:

**Reviewer 1**

**Remark 1:** There is no literature review about similar work in Morocco, while the results presented in the present manuscript can be already found in the following published papers

**Answer:** We have added a literature review. We compared the results of our simulations with those already published in the papers given by the reviewer. Please see page 15.

**Remark 2:** Section 4.1 is not really a "result" section, it is rather a presentation of the study area and data

**Answer:** This has been corrected. Please see page 11.

**Remark 3:** Before using TRMM precipitation, a validation against observed data would be welcome. Moreover, no validation of the RCM simulations is provided with observations.

**Answer:** There is a validation between TRMM precipitation and observed data in Figures 6,7,8.

**Remark 4**: The figures are not clear, such as figures 7,8,9,10.

**Answer**: The quality of the figures has been improved. Please see pages 12, 13, 14 and 16.

**Remark 5**: An interesting contribution of this article would be to provide an assessment of changes in runoff sensitivity coefficients (to precipitation and evapotranspiration changes), for both the historical periods and the RCPs.

**Answer**: Although this is an interest path, its feasibility is highly questionable because the RCM fails to predict runoff correctly especially for a rapidly changing topography case.

**Reviewer 2**

**Remark 1**: I think the simplicity also necessitates examination of the uncertainties in the approach and results. Where possible the uncertainty should be directly quantified. Where that is not possible, it should be discussed. For example, the authors conclude that the elasticity coefficient is near 1.6 due to the similarity of the three regression slopes in Figure 10. However, each regression slope has significant uncertainty in v its value due to the limited number of observations in each plot. The regression uncertainty is easy to quantify because the authors are using standard regression analysis. It would also be interesting to compare the implied range of elasticity from the historical data to the range inferred from the empirical relationships. Similarly, summarizing the results with single lines in Figure 12 seems to overstate the certainty of the analysis. Could ranges be presented instead?

Answer : Uncertainty quantification is surely a relevant question that should be addressed, especially in the context of climate change impact on the conception of hydraulic structures such as the Water Highway. As suggested by the reviewer, uncertainty has been addressed in the revised manuscript (see page 17). Figure 12 has also been adjusted according to this analysis (see page 20).

**Remark 2:** The analysis is largely based on the assumption that dry years in the past produced runoff in the same manner as a dry climate will. An analogous assumption is made for PET changes. I think

the paper would be strengthened by a short discussion about factors that may limit the applicability of this approach/assumption (or why this is a good assumption). For example, this approach seems to imply that the within-year variations of rainfall in the dry climate will be similar to the within-year variations of rainfall during dry years. I think it would be helpful to discuss such implicit assumptions (and to evaluate them if possible). It seems like sufficient data are available to investigate the seasonality assumption. It also seems to imply that the vegetation cover changes the same way in response to a climate change as to a single year drought. Can that assumption be justified? Further- more, the analysis seems to assume independence between the effects of PET and precipitation variables. Such independence would be violated if there is a transition from snowfall to rainfall. Would that confound the analysis? Does the historical record suggest such independence? Is such a transition expected?

Answer: Every model is built following some unavoidable assumptions. Given the availability of data we had, we were constrained to use the method presented in the paper. However, the remark remains relevant, and an analysis of the seasonality of precipitation will be added to the manuscript together with some discussion about the assumptions. A discussion about the potential impact and limitations of our methodology regarding the results has been added in the corrected version of our Manuscript. We have also confronted our results with other published work using different methodologies in order to assess climate change impact on water availability (page 15). As far as a transition from snowfall to rainfall assumptions is concerned, the studied area receives barely some snowfall such that its role can be neglected in the hydrological cycle.

**Remark 3:** The analysis makes an implicit steady-state assumption when evaluating the water highway project. Specifically, forecasts of 31 years into the future are used to analyse the project's viability, but those conditions may not be representative of the project design life. Imagine if the design life is 30 years. If built today, the project would experience a range of conditions between the current climate and the predicted climate that might make it more viable. The authors should mention this issue in their paper and provide an argument about why 31 years is appropriate for this assessment.

**Answer:** There is mainly two reasons why the 31 years is appropriate for this assessment. The first one is linked to climate change. In fact, this project has been designed in order to face future climate change impacts on the southern Moroccan areas. If the project may jeopardize the water security of the northern region how can it face the southern region water demand as well? The second reason is purely from a feasibility perspective. The project will need more than 10 years to be fully built, not to mention that the project has not started yet (The construction was planned to start in 2018). In addition, the Moroccan government has only made available the budget for the first phase. The second and the third phase are still waiting for the necessary funds, which will certainly delay the project for another 10 years. This discussion has been added in the introduction.

**Remark 4:** The analysis relies on dynamically downscaled GCM results. However, the paper does not discuss how that downscaling was accomplished and only cites another paper that is in preparation. I find this problematic since the validity of that work has not been substantiated by a peer-reviewed publication.

**Answer:** A manuscript of the companion paper will be included as supplementary information. Moreover, we confronted the results of our simulations to other published works (please see page 15).

Specific Comments: All the specific comments were added to the revised manuscript.

---

## Author Response (AR2)

**Answer to the Editor**

Dear editor

We sincerely thank you for the positive feedback regarding our paper. We made sure to take into account all the provided remarks. The corrections were added in a red color to differentiate it from the previous correction. Furthermore, we corrected Figure 11 accordingly.

Best regards,

The authors.